# A Parallel Classification Model for Marine Mammal Sounds Based on Multi-Dimensional Feature Extraction and Data Augmentation

**DOI:** 10.3390/s22197443

**Published:** 2022-09-30

**Authors:** Wenyu Cai, Jifeng Zhu, Meiyan Zhang, Yong Yang

**Affiliations:** 1College of Electronics and Information, Hangzhou Dianzi University, Hangzhou 310018, China; 2College of Electrical Engineering, Zhejiang University of Water Resources and Electric Power, Hangzhou 310018, China

**Keywords:** feature extraction, convolutional neural network, transfer learning, data augmentation, marine mammal classification

## Abstract

Due to the poor visibility of the deep-sea environment, acoustic signals are often collected and analyzed to explore the behavior of marine species. With the progress of underwater signal-acquisition technology, the amount of acoustic data obtained from the ocean has exceeded the limit that human can process manually, so designing efficient marine-mammal classification algorithms has become a research hotspot. In this paper, we design a classification model based on a multi-channel parallel structure, which can process multi-dimensional acoustic features extracted from audio samples, and fuse the prediction results of different channels through a trainable full connection layer. It uses transfer learning to obtain faster convergence speed, and introduces data augmentation to improve the classification accuracy. The k-fold cross-validation method was used to segment the data set to comprehensively evaluate the prediction accuracy and robustness of the model. The evaluation results showed that the model can achieve a mean accuracy of 95.21% while maintaining a standard deviation of 0.65%. There was excellent consistency in performance over multiple tests.

## 1. Introduction

Human activities in the oceans have a huge impact on marine ecological environments. In particular, noises generated by oil-exploration platforms, large offshore wind-power equipment, and ships are serious threats to the survival of marine animals. Therefore, the study of the quantitative distribution and migration behavior of these animals is of great significance in maintaining biodiversity and ecological balance [1,2,3,4]. As one of the most important species in the ocean, marine mammals are highly sensitive to changes in the environmental in which they live, so they are often regarded as the most suitable research objects to evaluate the quality of the ecological environment [5]. Marine mammals can generally be divided into five categories in the field of biology, namely cetaceans, pinnipeds, sirenians, marine otters, and polar bears [6].

Marine mammals spend most of their life under the water surface and rely on acoustic signals to communicate with each other. The most effective way to collect their acoustic signals is to use passive acoustic monitoring (PAM) that can cover a wide range of the target sea areas [7]. Since this type of system does not emit sonar signals, it can protect the living habits of marine animals to the maximum extent. Since the PAM system is located underwater, detection of behavior is not affected by bad weather, so it has a strong stability. The PAM system generally consists of an array of hydrophones with different depths. It can be mounted on a moving ship by cables to form a mobile PAM, which can provide continuous and real-time monitoring data, or it can form a fixed PAM by being connected to buoys or anchored at the sea bottom, which can provide long-term automatic offline data acquisition [8]. A typical PAM equipment is the underwater acoustic signal acquisition system composed of 160 hydrophones arranged in the Norwegian Sea, which has a coverage of 100,000 km^2^ [9]. Figure 1 shows an audio signal of about 6 s collected from Antarctica by PAM equipment [10]. The bottom half of the figure is the unprocessed time-domain representation of the audio signal, which is sufficient to distinguish the quiet state from the target state. In order to further analyze the audio sample, the signal can be converted from the time domain to the time–frequency (T–F) domain through short-time Fourier transform, where the target marine mammal sound pattern can be seen. However, the spectrogram needs to be further processed before the next classification algorithm, because the audio is often filled with long-term quiet data and the simultaneous existence of multiple species. As the audio represented in this figure shows, most of the time there is no target signal and only ambient noise, and the signal came from the call of a Weddell seal that lives in Antarctica. We need to use an algorithm to remove the quiet state and filter out the desired target from the noisy background in the preprocessing stage.

After collecting a large amount of acoustic data, target samples can be identified by a special analysis method. Traditionally, the task is performed manually by experts in the field of underwater acoustics who listen carefully to the audio or observe the spectrum to identify whether there are acoustic signals of target animals. However, this method has some limitations. One limitation is that humans can only hear sounds between 20 Hz and 20 kHz, but the sound of many marine mammals exceeds this range. For example, narwhals captured by researchers in Northwest Greenland had a call signal of more than 200 kHz, hence part of their sounds cannot be heard by human ears [11].

A natural idea is to design an end-to-end algorithm without any manual processing. This has the advantage of not using domain-knowledge-based feature engineering. The goal of some related research is to enable the algorithm to automatically extract features from raw waves, and this technical path is called feature learning. Its representative methods include dictionary learning [12], sparse representation learning [13], and deep learning [14]. Some researchers believe that features based on feature engineering are still more efficient than end-to-end models based on feature learning in some cases [15]. Typical examples of extraction from the time domain are zero crossing rate [16], autocorrelation coefficients [17], and the time-domain envelope [18]. Compared with the time-domain features, there are more optional features in the frequency domain, such as spectral envelope [19], spectral flux [20], and Mel spectrogram [21]. By decomposing these features, we can get cepstral features, Mel frequency cepstral coefficients (MFCCs) [22], linear-frequency cepstral coefficients (LFCCs) [23], gammatone-feature cepstral coefficients [24], and linear predictive coding (LPC) [25], as well as some feature histograms of oriented gradients [26], sub-band power distribution [27], and local binary patterns [28] developed from the field of image recognition.

Several studies have proved that in most cases, the performance of frequency-domain features is better than that of time-domain features in target detection and classification tasks [29]. There is some interesting evidence about this in the evolutionary history of biology. In the human cochlear structure, which has been formed through evolution, the auditory system responds to the external sound with loudness and pitch, which corresponds to the power and frequency of the sound, respectively, and the frequency-domain feature spectrogram contains all these information. The equal loudness contours and critical-band theory discovered by Harvey Fletcher in 1933 [30] proved that the relationship between critical band and sound frequency is a nonlinear relationship and further simplified the spectrogram. Through the matrix operation of the spectrogram and nonlinear Mel-filter banks, the widely used acoustic features Mel spectrogram is obtained.

After obtaining the features, we need to find an efficient classification algorithm to automatically identify the audio. Before machine learning became popular, most of the research was based on traditional pattern-recognition algorithms, such as spectrogram correlation [31], edge detection [32], dynamic time-warping [33] and energy-ratio mapping [34]. Compared to these traditional recognition algorithms, machine learning based on a large number of data samples can fit a very complex and accurate prediction function, so it is increasingly used in the field of acoustic detection. Typical machine learning based audio recognition algorithms comprises decision trees [35], linear discriminant analysis (LDA) [36], support vector machines (SVMs) [37], the Gaussian mixture model (GMM) [38], self-organizing maps (SOMs) [39], long short-term memory (LSTM) [40], the hidden Markov model (HMM) [41], and convolutional neural networks (CNNs) [42,43,44,45,46]. As computers become more powerful, researchers are processing larger amounts of data. Therefore, CNN-based algorithms, due to their data-driven attributes, can find the complex relationship between features and labels from a large number of samples, thus showing more potential than other algorithms in the application field [47].The mechanism of CNNs is to extract local features from the low-level structure, and then the high-level structure generates global judgment. This process can be understood through visualization of the receptive field. Figure 2 is a marine mammal audio selected at random from the Watkins Marine Mammal Sound Database [10], and fed into the pre-trained CNN model Xception after extracting the spectrogram [48]. The receptive field is visualized at layers 6, 12, and 48. It can be seen that some local features similar to the input spectrogram can be distinguished at the lower level, but the meaning of the receptive field cannot be understood by human eyes at the higher level.

In the field of marine-mammal sound classification, some research has been carried out based on machine-learning algorithms. For example, González-Hernández et al. [49] used the octave analysis method to extract the features of underwater acoustic signals, and constructed a fully connected neural network to classify 11 types of underwater sounds, including several marine mammal species, and achieved a classification accuracy of about 90% under the condition of low computational load. Kirsebomet et al. [36] achieved a recall ratio of 80% and a precision of 90% for the North Atlantic right whale under diverse environmental conditions through two-dimensional CNN, and proved that dataset with higher variance was beneficial to the model’s training. Luet et al. [43] solved the common overfitting problem of deep models by modifying the last layers of a pre-trained model, and performed excellent detection and classification of marine-mammal calls with extensive overlapping living areas. The model achieved 97.42% classification accuracy in a dataset with a high signal-to-noise ratio. Frasier et al. [50] proposed an algorithm to classify dolphin click signals, it used time–frequency profiles and click-interval distributions as features, and developed an automatic classification method based on unsupervised networks. The majority of current studies are based on single-dimensional acoustic features [51], but the complexity of acoustic signals means that a single acoustic feature cannot describe the signal well. Better performance can be achieved by selecting and fusing acoustic features from different dimensions.

Research of this paper was focused on cetaceans. These can be divided into odontocetes and mysticetes [52,53]. In order to improve the accuracy of marine-mammal sound classification, this paper proposes a classification model named MDF-PNet; MDF refers to multi-dimensional features, respectively, and P refers to parallel. Its innovation is in the design of a multi-channel parallel model, which can fuse four different and complementary acoustic features, and use a trainable full-connection layer to get the final prediction result. By using transfer-learning techniques, it can achieve faster convergence speed. At the same time, through data augmentation of the data set, the mean accuracy is further improved. Using the k-fold cross validation strategy to cut the data set, we can evaluate the accuracy and robustness of the model.

## 2. Materials and Methods

### 2.1. Data Description

The quality of the acoustic dataset plays an important role on machine-learning models. A high signal-to-noise ratio (SNR) and accurately labeled dataset are beneficial in ensuring more accurate parameters and higher prediction accuracy of unknown samples. The data used in this paper is from the Watkins Marine Mammal Sound Database [10,54] and contains more than 15,000 labeled digital sound clips of 51 marine mammal species from the 1940s to the 2000s. The location of the sampling area extends from Bering Strait to the coast of Australia, and the original sampling rate ranged from 5.12k to 192k.

This paper is dedicated to classifying six species in ‘all cuts’ to evaluate the proposed model. Considering that too few samples will lead to bias in test results, the species were selected based on sample size to better measure the performance of the model. The specific distribution and characteristics of these samples are shown in Table 1.

By randomly selecting the audio sample for each species and performing STFT (short-time Fourier transform) on the sample, the corresponding relationship between the original audio waveform and the spectrogram could be obtained as Figure 3. It can be seen that although each audio looks very similar in the time domain, the difference between them becomes very obvious after the signals are transformed from the time domain to the frequency domain. Therefore, the features used in this paper mainly come from the frequency domain.

### 2.2. MDF-PNet Structure

The detailed structure of MDF-PNet model is shown in Figure 4. The MDF-PNet model consists of four neural network branches based on transfer learning: Mel spectrogram branch, MFCC branch, LFCC branch, and mean MFCC branch. Each branch processes different kinds of features, and four special acoustic features were chosen since they can provide complementary feature descriptions from different perspectives. Specifically, the Mel spectrogram describes the samples from an image visual perspective, while MFCC and LFCC use statistical methods to extract the essential information of the samples. Among these branches, the Mel spectrogram branch is a pre-trained DenseNet [55], and the shape of its input feature is 128 × 345 × 3 (frequency × time × channel). The MFCC branch and LFCC branch are pre-trained MobileNet [56] and Xception [48], respectively, and the input tensors are also processed in the same way as the Mel spectrogram branch. In addition, the input of the mean MFCC branch is a one-dimensional tensor. The last layer of four branches is the softmax layer with six nodes, which corresponds to six kinds of marine-mammal species, and its output tensor represents the probability that the sample belongs to each species. The calculation formula of softmax layer output is described in Equation (1):(1)$$y_{j}=\frac{e^{x_{j}}}{\sum_{i=1}^{N}e^{x_{i}}}\;\;\;1⩽j⩽6$$
where *x**_i_* is the output value of the *i*th node of fully connected layer, and *y_j_* is the output value of the *j*th node of softmax layer. The parameter *N* is 6, which means that each branch will eventually divides all samples into six categories and it predicts a vector with the shape of [6 × 1].

The MDF-PNet uses stochastic gradient descent (SGD) as the optimization method of parameters. Compared to other optimization methods, SGD has a higher parameter-update frequency so as to reduce the probability of jumping into local optimal solutions. The cross-entropy (CE) loss is used as the optimization target for the proposed model, which is easier to train than mean-square-error (MSE) loss [57]. In this paper, the cross-entropy loss is applied to evaluate the difference between predicted and actual labels. The calculation method of cross-entropy loss is shown as Equation (2):
(2)$$L=-\frac{1}{N}\sum_{i=1}^{N}\sum_{j=1}^{M}y_{ij}\log\left(p_{ij}\right)$$
where *L* represents the cross-entropy loss of a batch of samples, *N* is the number of samples in this batch, and *M* is the number of species in classification. *y_ij_* refers to the *j*th elements of one-hot encoded label of the *i*th sample and *p_ij_* refers to the prediction probability that the *i*th sample belongs to category *j*. *L* is the optimization target and by observing the loss curve in the process of model training, we can see that the value of *L* will gradually decrease, which means that the model has gradually fitted the correct mapping curve.

The predicted results of the four branches are denoted as Y1, Y2, Y3, and Y4, respectively. The four independent prediction results are connected to form a [24 × 1] vector, and then a final [6 × 1] vector is obtained through a trainable fully connected layer and softmax layer as the prediction result. Compared with the fusion method of averaging each branch in some existing studies [58], the learnable weight parameters have better flexibility. As shown in Figure 5, this fusion layer is named the dense layer, which has better adjustability for the weight of each branch compared with the common average calculation method.

### 2.3. Data Augmentation Method

For the classification task in deep learning, one basic requirement is that the number of samples in different categories is similar since the data imbalance will affect the generalization ability of the classifier with fewer samples [59]. However, there were too few samples of the three species in the original dataset, so data augmentation was used to keep their sample sizes roughly the same. Traditional data augmentation methods include flipping, rotating, scaling, cropping, and color-jittering. Previous studies have proved that these methods can improve the classification accuracy [60]. For sound data, the samples obtained after flipping the waveform were opposite to the original samples on the time axis in the T–F domain, and belonged to different samples from the perspective of the classification model. Such data augmentation method is often used in sound classification scenes [61]. In this paper, the raw waveform was horizontally flipped for data augmentation and Figure 6 is an example of sample-flipping. The total number of samples after augmentation increased from 5514 to 7359, and the number of samples of all species was basically at the same level.

### 2.4. Feature Extraction Algorithm

Several studies have proposed that MFCC has stronger representation ability in the low-frequency region, while LFCC is better in the high-frequency region [62]. Therefore, we can improve the expressive ability by fusing different types of features and their unique advantages. Four frequency-domain features were selected in this paper: including Mel spectrogram, MFCC, LFCC, and mean MFCC. MFCC is the most widely used feature in the field of audio recognition. The specific calculation can be divided into the following two steps:

Step (1): Perform STFT on the original audio and filter through Mel filter banks. Since the acquisition equipment when collecting data has different sampling rates, this parameter represents the number of times the equipment samples the signal every second. Therefore, all samples should be resampled to a certain fixed frequency, which is 10 kHz in this paper, before the short-time Fourier transform. Although the shape of input tensor about the MDF-PNet model is fixed, the original length of each sample is inconsistent. Therefore, the audio samples are processed through the strategy of random cropping to get the same length. For samples that are too long, the cropping is carried out, and for the short sample, the value zero is randomly added on both sides. Therefore, the length of 30,000 sampling points was unified equivalent to 3 s of audio. After preprocessing, 1024 points were selected as the frame length and 512 points as the step size, and then STFT operation was performed on the audio sample to obtain a complex matrix *S_t_*(*k*) with size 59 × 513 (time × frequency). Then, the power spectrum of audio was obtained by power operation, which is real matrix [*S_t_*(*k*)]^2^ with size 59 × 513. Finally, according to the number of Mel filters selected, the Mel filter bank matrix *H_m_*(*k*) with size of 513 × 128 was generated and multiplied by the results of STFT to obtain a power spectrum of Mel scale, i.e., a 59 × 128 real matrix *X_m_*(*t*). The calculation process is shown in Equations (3) and (4) [63]:
(3)$$X_{m}\left(t\right)=\sum_{k=0}^{N-1}{\left[S_{t}\left(k\right)\right]}^{2}H_{m}\left(k\right),\;\;\;0⩽t<L,\;\;\;0⩽m<M$$
(4)$$H_{m}\left(k\right)=\{\begin{array}{ll} \frac{k\;-\;o\left(m\right)}{c\left(m\right)\;-\;o\left(m\right)} & \;o\left(m\right)⩽k⩽c\left(m\right)\\ \frac{h\left(m\right)\;-\;k}{h\left(m\right)\;-\;c\left(m\right)} & \;c\left(m\right)⩽k⩽h\left(m\right) \end{array}$$
where *X_m_*(*t*) denotes the power spectrum matrix, *L* denotes the number of data frames in the direction of time axis, *M* denotes the number of Mel filter banks, *S_t_*(*k*) denotes the spectrum obtained by STFT, *H_m_*(*k*) is the expression of Mel filter banks, where *o*(*m*), *c*(*m*), and *h*(*m*) are frequency points, and their interval determines the type of filter banks.

Step (2): Perform discrete cosine transform (DCT) on logarithm of power spectrum. Firstly, the power spectrum is converted into decibels (dB unit), and then DCT transformation is carried out to obtain the MFCC matrix. The dimension of MFCC matrix is independent of the number of Mel filter banks in the previous step, but determined by the cepstral coefficient. Herein, the cepstral coefficient was 32, a matrix with size 59 × 32 was finally obtained by the calculation of Equation (5).
(5)$$\mathrm{mfcc}\left(t,c\right)=\sum_{m=1}^{M}\log\left(X_{m}\left(t\right)\right)\cos\left(\frac{c\pi \left(m-0.5\right)}{M}\right)$$
where mfcc(*t*,*c*) is the calculated MFCC matrix, *t* denotes the index in the time axis, *c* denotes the index of cepstral coefficient, *M* denotes the number of Mel filter banks, and *X_m_*(*t*) denotes the input power spectrum matrix. In some applications, the dynamic features of MFCC are also calculated, i.e., the first-order difference and the second-order difference form a high-dimensional feature together with the static features [64].

The calculation process of the Mel spectrogram is similar, except that it only uses Step (1). The difference between LFCC and MFCC is that the Mel filter bank in Step (1) is replaced by a linear filter bank for LFCC.

To summarize, the difference between these features can be intuitively observed through visualization. An audio sample was randomly selected from the data set in Table 1, and a high-resolution spectrogram was obtained through STFT to compare. After feature extraction processing through the above steps, four real number matrices could be obtained. The visualization results are shown in Figure 7, where the X-axis represents time with second unit and Y-axis represents frequency with Hz unit. It can be seen that although the resolution of Figure 7b is much lower than Figure 7a, the signal contour can still be seen. Furthermore, signals cannot be directly distinguished from Figure 7c,d after DCT transformation. In conclusion, it is promising to combine these four features to improve audio recognition accuracy.

### 2.5. Transfer-Learning Method

Three pre-training models including DenseNet-201, MobileNet, and Xception were used to initialize parameters in the proposed MDF-PNet, which were based on the subset of ImageNet and fully trained. The number of parameters of these networks were 20.2 M, 4.2 M, and 22.9 M, and the depths of these models were 201, 88, and 126 layers, respectively. Since the feature tensor generated in the feature extraction stage was different from the shape of ImageNet, each pre-training model needed to be modified slightly before usage. The method used was to modify the input layer of all models to match the corresponding feature tensor size and then remove the weight of full connection layer, and finally add the full connection layer consistent with the sample label dimension and retrain it.

Compared with the huge computational workload required by two-dimensional CNN, one-dimensional CNN could achieve the same excellent effect in a simpler way. As a part of the proposed MDF-PNet, it can improve the overall performance of classification. So far, in addition to these three pre-training models, a one-dimensional CNN as the fourth parallel structure was constructed to match the 1D Mean MFCC vector. After training 3000 epochs, the obtained weights were saved as the initial weights of the fourth model. The structure of this one-dimensional CNN is shown in Table 2.

### 2.6. Data Segmentation

In the field of statistics, it is of great significance to reasonably divide the data set so as to obtain a more balanced sample distribution. Especially for small-scale samples, simply dividing the data set into training sets and test sets will often lead to the inability to accurately evaluate the performance of the model. Compared with this segmentation method, a more reasonable way is to use k-fold cross validation to test the model repeatedly, and use the mean value to characterize the performance of the model. As shown in Figure 8 below, this method firstly cuts out 20% of the data as the test set, which is not involved in subsequent parameter adjustment and model training, and then divides the remaining 80% of the data into five parts for cycle training and testing to obtain five validation accuracy values. Finally, the prediction accuracy of the model is obtained by calculating the mean value. Accuracy value can be calculated by Equation (6) below [65]:(6)$$\mathrm{Accuracy}=\frac{1}{N}\sum_{i=1}^{N}\frac{TP_{i}+TN_{i}}{TP_{i}+TN_{i}+FP_{i}+FN_{i}}$$
where *N* refers to the number of species, the value is 6 in this paper. The metrics in multi-class classification are derived from binary classification. *TP*, *FP*, *TN*, and *FN* refer to true positive, false positive, true negative and false negative, respectively. Accuracy is an indicator designed to evaluate the consistency between the prediction result of the model and the label, so a more intuitive understanding can be considered as the ratio between the number of correctly classified samples and the total number of samples. The following test results for baseline and MDF-PNet were obtained based on this data segmentation method.

## 3. Results

### 3.1. Performance Comparison between Baseline and MDF-PNet

In order to compare the performance of different models in similar fields, it is necessary to choose recent research in the field of ecological informatics as the baseline. In reference [43], the authors proposed a modified AlexNet based on transfer learning to detect and classify marine-mammal sounds. The data used by the author come from the ‘best of cuts’ part of the marine mammal call dataset. Compared to the part marked ‘all cuts’, this part of dataset has a higher signal-to-noise ratio, so it can often obtain excellent classification accuracy. In this paper, the model proposed in this reference is used as the baseline and compared with the proposed MDF-PNet.

After the data set was segmented using the above data segmentation method, the performance of baseline and MDF-PNet on train set and validation set was tested. In order to maintain strict consistency in data distribution, the same random seed was used to segment the data sets. Loss curve and accuracy curve can accurately visualize this process. The former represents the output of loss function, while the latter reflects the accuracy of prediction. Considering that the generalization ability of the model on validation set is the most important index pursued by the algorithm, Figure 9 and Figure 10, respectively, represent the iterative process of baseline and MDF-PNet on the validation set. The light-colored lines in the figure represent the process of five cycles of testing, and the dark lines represent their mean values.

In addition to the mean, the standard deviation is also an important indicator in statistics. It can measure the dispersion of the results in the case of multiple measurements. A model with better performance should have a lower standard deviation. Table 3 shows the mean and standard deviation of baseline and MDF-PNet under the five tests.

In order to more intuitively compare the accuracy of the two models, the two models were used to predict the same batch of data for evaluation. This batch of data is the part of the test set obtained by the k-fold segmentation strategy above. Since it has never participated in the previous training, it is a batch of completely independent data. The predicted results are represented using the confusion matrix in Figure 11. The labels on the axes correspond to the abbreviation of species name, and their full names are shown in Table 1 above.

When evaluating the performance of binary classification models, precision, recall and F1 score can be used to comprehensively characterize the classification ability of the model. The specific steps are as follows: First, it is assumed that the labels of the samples can be divided into positive and negative. Then prediction results are compared with the real labels to obtain the four categories of true positive, true negative, false positive, and false negative. Finally, according to the different focus of the research objectives, the following statistical metrics can be obtained in Equations (7)–(9) [66]:(7)$$\mathrm{Precision}=\frac{\mathrm{True}\;\mathrm{Positive}}{\mathrm{True}\;\mathrm{Positive}\;+\;\mathrm{False}\;\mathrm{Positive}}$$
(8)$$\mathrm{Recall}=\frac{\mathrm{True}\;\mathrm{Positive}}{\mathrm{True}\;\mathrm{Positive}\;+\;\mathrm{False}\;\mathrm{Negative}}$$
(9)$$F1\;\mathrm{Score}=\frac{2\times \;\mathrm{Precision}\times \mathrm{Recall}}{\mathrm{Precision}\;+\;\mathrm{Recall}}$$

The focus of precision and recall in the above formulae is different. The former focuses on evaluating the accuracy of prediction results, while the latter focuses on measuring the coverage of prediction results. To some extent, these two indicators have certain contradictory attributes, that is, when only one indicator is considered to be highest, the other indicator will perform worse. So F1 score, as a weighted statistical index of precision and recall, can be used to evaluate the model performance more comprehensively. In the multi-label classification task, the above statistical metrics are still effective, so the confusion matrix can be further calculated by the above formula to obtain the statistical results shown in Table 4.

### 3.2. Effect of Data Augmentation on Performance

Finally, in order to study the effect of the data augmentation method on the performance of baseline and MDF-PNet, the following cross-test was conducted based on the model used and data augmentation was performed. In order to keep the consistency of the input data under different conditions, this step is also based on the above data segmentation method. The mean accuracy curve obtained by multiple tests on the validation set is shown in Figure 12. Table 5 lists the mean accuracy and standard deviation of the cross-test, where the better results are highlighted in red.

## 4. Discussion

By comparing Figure 9 and Figure 10, we can clearly see three advantages of MDF-PNet compared with baseline: (1) The convergence speed is faster. MDF-PNet only needed 20 epochs to make the loss curve become stable, while the baseline required about 200 epochs to achieve the same effect. (2) The mean accuracy of prediction was higher. When the results of five tests were averaged, the mean prediction accuracy of MDF-PNet for the validation set was 95.21%, while the mean prediction accuracy of baseline under the same test conditions was 88.40%. (3) The standard deviation of the prediction results was smaller. The standard deviation of MDF-PNet was 0.65% for five tests, while the baseline was 1.36%. This means that MDF-PNet has less dispersion and better consistency in its prediction performance. Table 3 records these statistical results in detail.

Figure 11 illustrates the performance of the baseline and MDF-PNet on the same batch of samples. It is obvious that MDF-PNet is better than baseline in the classification ability of most marine mammal species. Compared with the confusion matrix, by introducing statistical indicators to quantify the classification ability of them, we can get a more accurate perception of the improvement of the model performance. All indicators obtained according to Equations (6)–(8) have been listed in detail in Table 4. The left side of the slash in each cell belongs to baseline, and the corresponding statistics results on the right belong to MDF-PNet. The better results among them are marked with red. As can be seen, the performance of MDF-PNet was better than baseline in most metrics, which fully reflects the effective performance improvement of the model. There are many factors that affect the performance of a model. One of the most important is the quality and size of the dataset used. Therefore, the comparison of metrics is only valuable when dealing with the same dataset. Trawickiet et al. [41] used a hidden Markov model to classify marine mammals in the subset of the dataset, and obtained a classification accuracy of 84.11%, but too few samples were used, so the obtained results had a high variance. There is also work based on different dataset of marine-mammal calls. For example, Kirsebom et al. [36] used a deep neural network with skip structure to detect North Atlantic right whale upcalls. After the evaluation on the test set, their method could achieve a recall up to 80% and a precision of 90% on the dataset, both of which seem to be slightly worse than the model proposed in this paper.

Figure 12 shows the effect of the data augmentation step for baseline and MDF-PNet, in evaluating the significance of data equalization for the model performance, and comparison was performed under completely consistent data segmentation criteria. Since the convergence speed of baseline was slower than MDF-PNet, it had more iterations on coordinates than the latter.

The numerical results of Figure 12 are shown in Table 5. It can be seen that for baseline that the data-augmentation step increased its mean accuracy from 81.89% to 88.40%, an increase of 6.51%. For MDF-PNet, this step increased mean accuracy from 93.67% to 95.21%, an increase of 1.54%. For any model, this step makes the standard deviation of the five test results smaller and achieves better consistency, which fully demonstrates that this step is effective and necessary. The highest mean classification accuracy was achieved under MDF-PNet with data augmentation. Compared with baseline, its mean accuracy was increased by 6.81%.

## 5. Conclusions

In this paper, we propose a classification model for marine mammal sounds based on multi-dimensional feature extraction. The data augmentation step was applied to the unbalanced data to make the distribution more balanced among the species, thus improving the mean classification accuracy of the model. At the same time, MDF-PNet used transfer learning to obtain the initial weights and thus achieved faster convergence. In the stage of acoustic feature selection, the proposed MDF-PNet model used multiple complementary features in the frequency domain to combine their respective advantages, and a full connection layer was retrained at the back end to connect the prediction results of different branches.

After the data set was uniformly segmented by k-fold cross validation, the MDF-PNet and similar research in the same field were repeatedly compared and tested. It can be seen that the mean classification accuracy of MDF-PNet increased by 6.81% compared with the baseline, and there was a lower standard deviation between different test results, which indicates that the performance had been greatly improved.

In our future work, we will explore whether the proposed model has a better solution in the fusion stage of prediction results, such as retraining a deep model to replace the current trainable full connection layer.

## Figures and Tables

**Figure 1 sensors-22-07443-f001:**
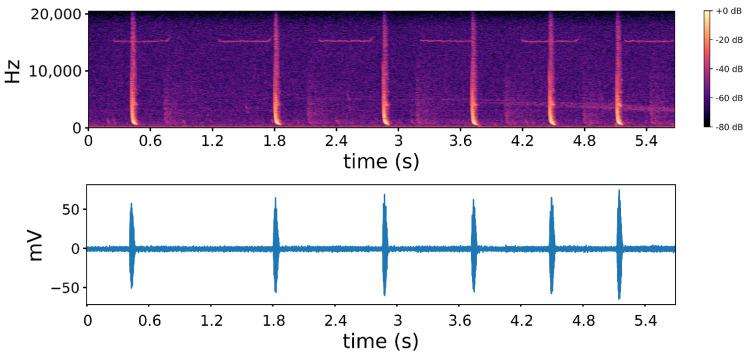
Time domain and T–F domain representation of an audio recording from Weddell seal.

**Figure 2 sensors-22-07443-f002:**
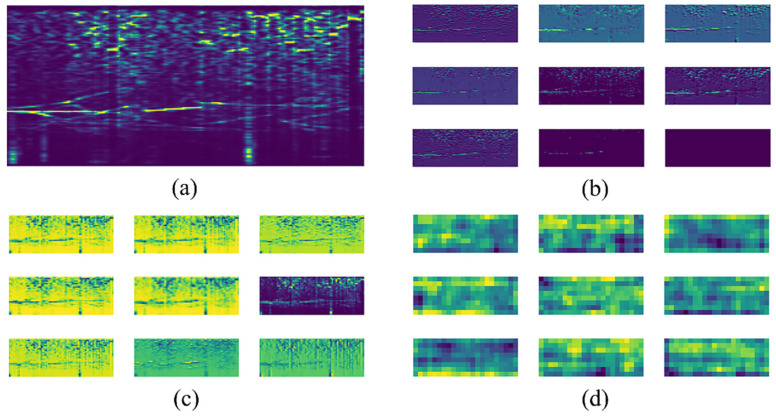
Visualization results on receptive fields of different convolution layers. (**a**) The spectrogram feature of the input, (**b**) the receptive field at layer 6, (**c**) the receptive field at layer 12, and (**d**) the receptive field at layer 48.

**Figure 3 sensors-22-07443-f003:**
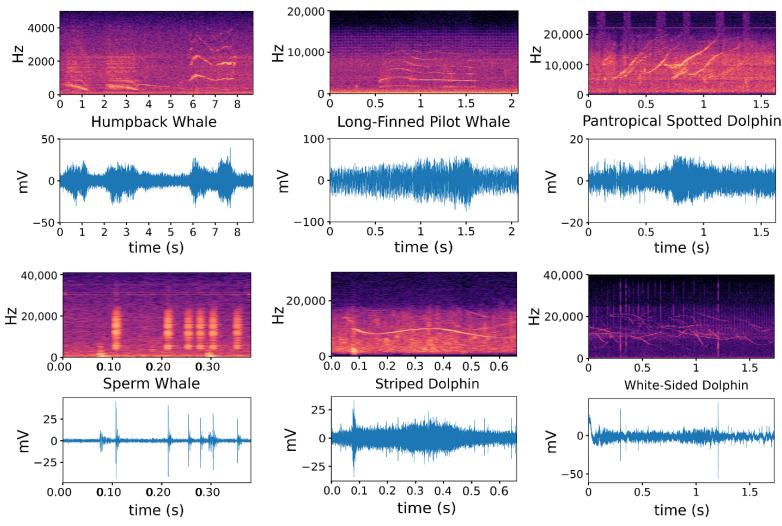
Comparison of acoustic signals from different species of marine mammals in time and frequency domains.

**Figure 4 sensors-22-07443-f004:**
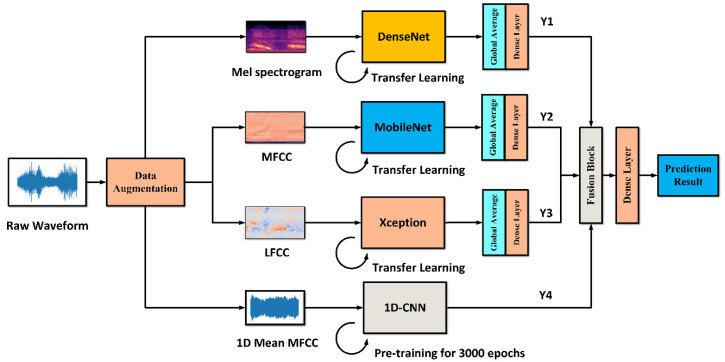
The network structure of MDF-PNet.

**Figure 5 sensors-22-07443-f005:**
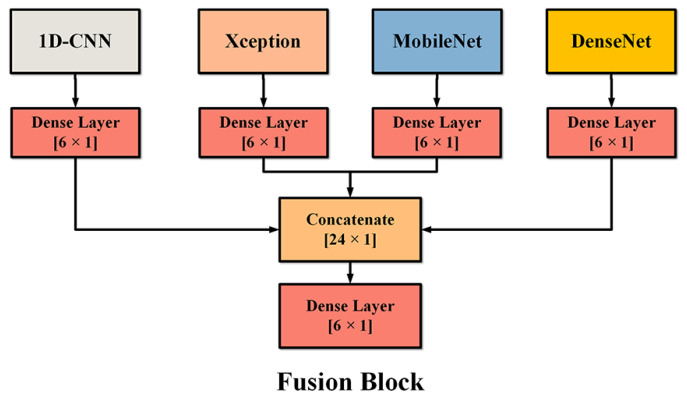
The fusion block of MDF-PNet at the connection layer.

**Figure 6 sensors-22-07443-f006:**
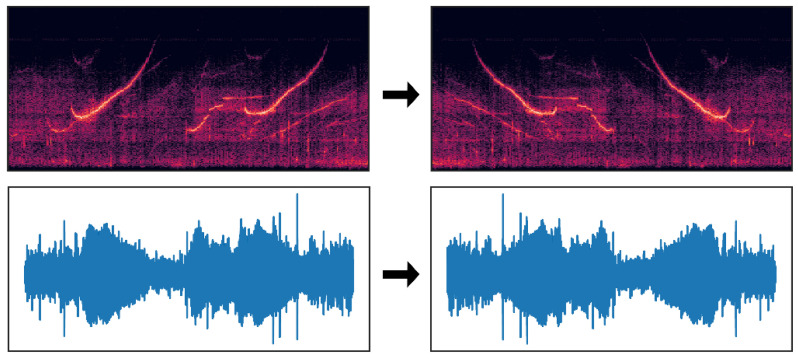
Data augmentation example.

**Figure 7 sensors-22-07443-f007:**
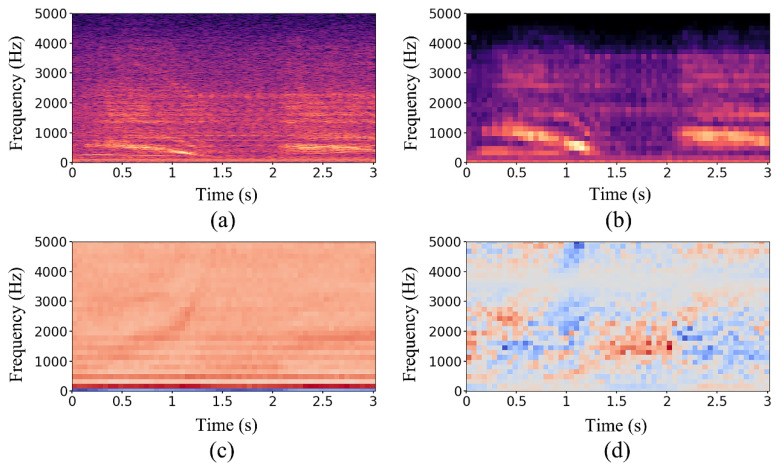
Frequency-domain features of the same audio sample. (**a**) Spectrogram (513 × 938), (**b**) Mel spectrogram (32 × 59), (**c**) MFCC (32 × 59), and (**d**) LFCC (32 × 59).

**Figure 8 sensors-22-07443-f008:**
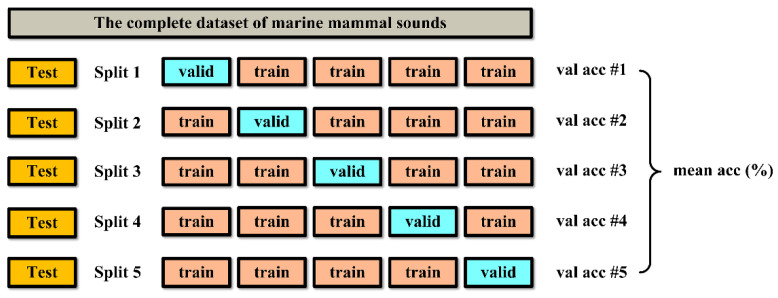
Segmentation method of data set.

**Figure 9 sensors-22-07443-f009:**
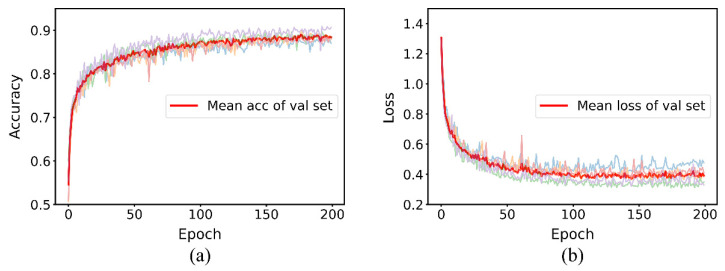
Five test results of baseline. (**a**) The accuracy curve and (**b**) the loss curve.

**Figure 10 sensors-22-07443-f010:**
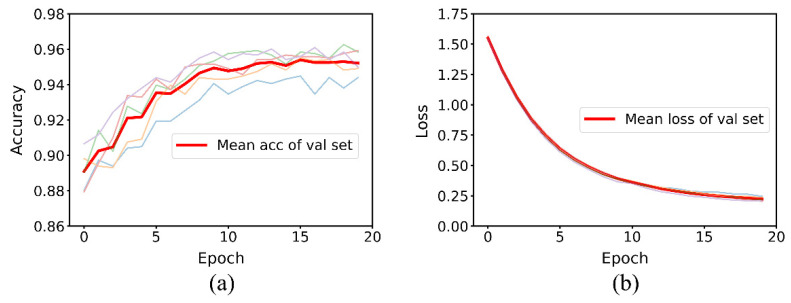
Five test results of MDF-PNet. (**a**) The accuracy curve and (**b**) the loss curve.

**Figure 11 sensors-22-07443-f011:**
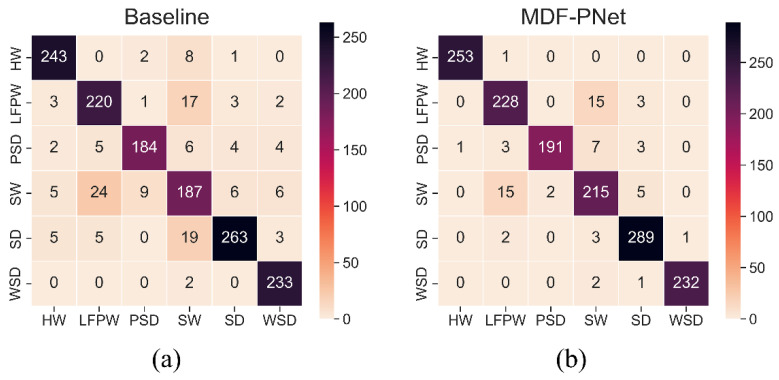
Confusion matrix of baseline and MDF-PNet on the same batch of samples. (**a**) Baseline and (**b**) MDF-PNet.

**Figure 12 sensors-22-07443-f012:**
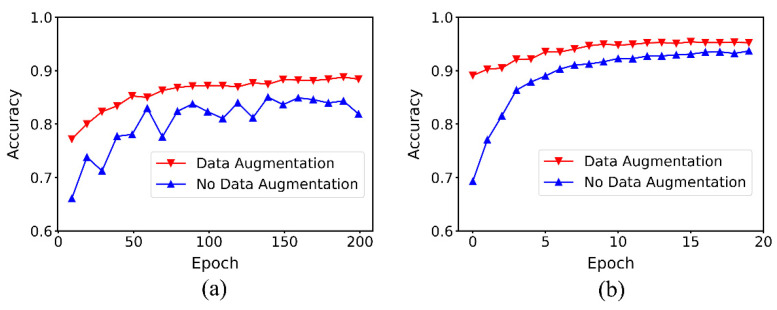
Cross-test the combination of two conditions. (**a**) baseline and (**b**) MDF-PNet.

**Table 1 sensors-22-07443-t001:** Distribution and characteristics of samples.

Species Name	Abbreviation	Sample Size (No Augmentation)	Sample Size (Augmentation)	Sampling Rate (Hz)
Humpback whale	HW	604	1208	5.12k–30k
Long-finned pilot whale	LFPW	1213	1213	23.5k–166.6k
Pantropical spotted dolphin	PSD	1034	1034	40.96k–192k
Sperm whale	SW	1422	1422	10k–166.6k
Striped dolphin	SD	681	1362	47.6k–192k
White-sided dolphin	WSD	560	1120	60.6k–166.6k
Total	-	5514	7359	5.12k–192k

**Table 2 sensors-22-07443-t002:** The detailed parameters of used 1D-CNN.

Layers	Description
Input layer	Input tensor: 128 × 1
{Conv + dropout + maxPooling1D} × 4	{Convolution: 16 kernels (3 × 1, stride = 1) Pooling: (2 × 1, stride = 1) Activation function: ReLu} × 4
Flatten + dropout	Convert the shape of tensor from 8 × 16 to 128 × 1
fully-connected	Units = 128
Fully-connected	Units = 6
Softmax Layer	Output tensor: 6 × 1

**Table 3 sensors-22-07443-t003:** Comparison of mean accuracy and standard deviation of the five tests.

Model/Data	Mean Accuracy of Five Tests	Standard Deviation
Baseline model	88.40%	1.36%
MDF-PNet model	95.21%	0.65%

**Table 4 sensors-22-07443-t004:** Comparison of statistical metrics performance between baseline and MDF-PNet.

Species/Metrics	Precision (Baseline/MDF-PNet) (%)	Recall (Baseline/MDF-PNet) (%)	F1 Score (Baseline/MDF-PNet) (%)
HW	94.19/99.61	95.67/99.61	94.92/99.61
LFPW	86.61/91.57	89.43/92.68	88.00/92.12
PSD	93.88/98.96	89.76/93.17	91.77/95.98
SW	78.24/88.84	78.90/90.72	78.57/89.77
SD	94.95/96.01	89.15/97.97	91.96/96.97
WSD	93.95/99.57	99.15/98.72	96.48/99.14

**Table 5 sensors-22-07443-t005:** Comparison of accuracy after cross-test.

Model	Mean Accuracy of Five Tests	Standard Deviation
Baseline + no data augmentation	81.89%	3.29%
Baseline + data augmentation	88.40%	1.36%
MDF-PNet + no data augmentation	93.67%	1.07%
MDF-PNet + data augmentation	95.21%	0.65%

## Data Availability

Not applicable.

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
