# Peer review of "A Parallel Classification Model for Marine Mammal Sounds Based on Multi-Dimensional Feature Extraction and Data Augmentation"

_sensors, 2022, doi:10.3390/s22197443_

Round 1

Reviewer 2 Report

The paper presents results of Machine Learning application for marine mammal sound classification. There are a lot of publications on this matter, and I would like to see clear explanation of the suggested method advantages in comparison with the other publications. For example, I can point out to several similar publications:

Shiu, Y., Palmer, K.J., Roch, M.A., Fleishman, E., Liu, X., Nosal, E.M., Helble, T., Cholewiak, D., Gillespie, D. and Klinck, H., 2020. Deep neural networks for automated detection of marine mammal species. Scientific reports10(1), pp.1-12.

Dugan, P.J., Clark, C.W., LeCun, Y.A. and Van Parijs, S.M., 2015. DCL System Using Deep Learning Approaches for Land-Based or Ship-Based Real Time Recognition and Localization of Marine Mammals. Bioacoustics Research Program, Cornell University Ithaca United States.

Zhong, M., Castellote, M., Dodhia, R., Lavista Ferres, J., Keogh, M. and Brewer, A., 2020. Beluga whale acoustic signal classification using deep learning neural network models. The Journal of the Acoustical Society of America147(3), pp.1834-1841.

Kirsebom, O.S., Frazao, F., Simard, Y., Roy, N., Matwin, S. and Giard, S., 2020. Performance of a deep neural network at detecting North Atlantic right whale upcalls. The Journal of the Acoustical Society of America147(4), pp.2636-2646.

Usman, A.M., Ogundile, O.O. and Versfeld, D.J., 2020. Review of automatic detection and classification techniques for cetacean vocalization. IEEE Access8, pp.105181-105206.

Allen, A.N., Harvey, M., Harrell, L., Jansen, A., Merkens, K.P., Wall, C.C., Cattiau, J. and Oleson, E.M., 2021. A convolutional neural network for automated detection of humpback whale song in a diverse, long-term passive acoustic dataset. Frontiers in Marine Science8, p.607321.

I also recommend to refer the review paper Bianco, M.J., Gerstoft, P., Traer, J., Ozanich, E., Roch, M.A., Gannot, S. and Deledalle, C.A., 2019. Machine learning in acoustics: Theory and applications. The Journal of the Acoustical Society of America146(5), pp.3590-3628.

 Several more comments:

1.    Authors are called their method MDF-PNet, but this abbreviation is not explained. I suggest not to use it the title and to explain why did they call this method MDF-PNet.

2.    Show scales for axes in Figure 3.

3.    Please explain how were the features extracted from the samples with various sampling rates? How were the samples transformed to the standard form?

4.     I am not sure that the data augmentation by the flipping of waveform may be used for increasing number of samples. I think that this method does not change frequency-domain features. Could you please calculate features for few direct and flipped samples.

Round 2

Reviewer 1 Report

The MS can be accepted

Author Response

Thank you!

Reviewer 2 Report

I am more or less satisfied by the majority of the paper corrections, but the authors did not provide proof that the data augmentation by the flipping of waveform can be used for increasing number of samples. The paper states that “the features used in this paper mainly come from the frequency domain”. The waveform flipping does not change the signal spectrum. It means that the spectral features are not changed after flipping and the data augmentation presented the paper has no sense.

I recommend to the authors to remove the data augmentation from the paper or to explain clearly why the waveform clipping can be used there?  

Author Response

Thank you!
